# The best features of diamond nanothread for nanofibre applications

Haifei Zhan[1,2], Gang Zhang[3], Vincent B.C. Tan[4] & Yuantong Gu[1]

Carbon fibres have attracted interest from both the scientific and engineering communities due to their outstanding physical properties. Here we report that recently synthesized ultrathin diamond nanothread not only possesses excellent torsional deformation capability, but also excellent interfacial load-transfer efficiency. Compared with (10,10) carbon nanotube bundles, the flattening of nanotubes is not observed in diamond nanothread bundles, which leads to a high-torsional elastic limit that is almost three times higher. Pull-out tests reveal that the diamond nanothread bundle has an interface transfer load of more than twice that of the carbon nanotube bundle, corresponding to an order of magnitude higher in terms of the interfacial shear strength. Such high load-transfer efficiency is attributed to the strong mechanical interlocking effect at the interface. These intriguing features suggest that diamond nanothread could be an excellent candidate for constructing next-generation carbon fibres.

[1] School of Chemistry, Physics and Mechanical Engineering, Science and Engineering Faculty, Queensland University of Technology (QUT), Brisbane, Queensland 4001, Australia. [2] School of Computing, Engineering and Mathematics, Western Sydney University, Locked Bag 1797, Penrith, New South Wales 2751, Australia. [3] Institute of High Performance Computing, Agency for Science, Technology and Research, 1 Fusionopolis Way, Singapore 138632, Singapore. [4] Department of Mechanical Engineering, National University of Singapore, 9 Engineering Drive 1, Singapore 11576, Singapore. Correspondence and requests for materials should be addressed to G.Z. (email: zhangg@ihpc.a-star.edu.sg) or to Y.G. (email: yuantong.gu@qut.edu.au).

Carbon nanotube (CNT) fibres have emerged as potential multifunctional nano-textiles in recent years. They have outstanding mechanical, chemical and physical properties that out-perform traditional carbon and polymeric fibres[1–3]. Various appealing applications have been proposed for CNT fibres, such as next-generation power transmission lines, twist-spun yarn-based artificial muscles which can respond to electrical, chemical or photonic excitations[4,5], aerospace electronics and field emission[1], batteries[6], intelligent textiles and structural composites[7].

In basic terms, CNT fibres comprise axially aligned and highly packed CNTs, which can be fabricated through either spinning[8,9] or twisting/rolling techniques[10]. Depending on the applied techniques, different CNT fibre structures have been reported such as knitted, parallel and twisted structures. Due to the complexity of their structures and different post-treatments processes, a large variation is found in their mechanical performance, for example, the strength ranges from 0.23 to 9.0 GPa, and the modulus ranges from 70 to 350 GPa (ref. 7). Additionally, for CNTs with large diameters, such as the most abundant (10,10) and (18,0) CNTs, the corresponding CNT bundle/rope will experience different metastable states due to the flattening of the constituent CNTs[11,12]. Flattening of the CNTs deteriorate the mechanical performance of the fibre. In this regard, increasing efforts have been attributed to optimize the manufacturing techniques to fabricate CNT fibres with desirable and controllable mechanical properties.

Very recently, a new type of ultrathin one-dimensional carbon nanostructure—diamond nanothread (DNT), was synthesized through a solid-state reaction of benzene under high pressure[13]. Similar to the hydrogenated (3,0) CNT, DNT has a hollow tubular structure, which is interrupted by Stone–Wales (SW) transformation defects. Encouraged by this experimental success, researchers found that there are different kinds of stable DNT structures through first principle calculations[14,15]. Preliminary studies have shown that the DNTs with SW transformation defects have excellent mechanical properties[16,17], a high stiffness of about 850 GPa, and a large bending rigidity of about $5.35 \times 10^{-28}$ N m[2]. Our following work revealed that the brittleness of DNTs can be changed via controlling the density of the SW transformation defects[18]. With these excellent mechanical properties, ultrathin dimensions and a non-smooth surface (compared to CNT), it is of interest to determine how DNTs may be used in fibre applications. For instance, what is the torsional property of DNT and how is load transferred in the DNT bundle structure?

To this end, here we carry out a series of *in silico* studies to explore the torsional characteristics of DNT bundles and their load-transfer efficiency. We find that DNTs are an ideal candidate for fibre applications. Not only do they possess excellent torsional deformation capability, they also possess excellent interfacial load-transfer efficiency.

## Results

**Twist deformation.** A DNT model was established based on recent experimental results and first principle calculations[13,14]. As illustrated in Fig. 1a, the DNT contains SW transformation defects (SWDs). Before constructing the DNT bundle, we investigated the equilibrium distances between two DNT strands through a series of relaxation simulations under isothermal-isobaric ensemble. In detail, two DNT models with different initial distances in lateral and longitudinal directions, and different orientations were examined (Fig. 1b–d). It is found that if the initial distance is smaller than $\sim 11$ Å, the two DNTs will attract each other (by the van der Waals force) leading to an equilibrium distance of $\sim 6.25$ Å. The final optimized geometry is

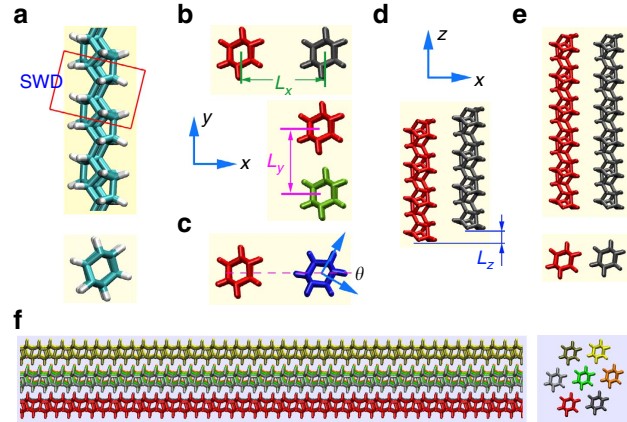

**Figure 1 | Schematic view of the diamond nanothread models.** (**a**) A sample DNT model with SW transformation defect (SWD), the top of the panel shows the side view and the bottom of the panel is the cross-sectional view. (**b**–**d**) Schematic views of two DNT models with different lateral distance ($L_x$ and $L_y$), orientations $\theta$ and longitudinal distance $L_z$. (**e**) The equilibrium morphology of two DNT strands obtained from minimization and relaxation simulation (under 1 K, with periodic boundary condition in length direction). (**f**) Schematic side view (left of panel) and cross-sectional view (right of panel) of a seven strand fibre made from DNTs.

uniform for all examined models. A typical fibre (length of $\sim 17$ nm) with seven strands was constructed as shown in Fig. 1f.

Figure 2a shows the gravimetric deformation energy ($J = \Delta E_t/m$) of the DNT bundle as a function of the twist rate ($\varepsilon_{t0} = \varphi/l_0$). Here, $\Delta E_t$, $m$, $\varphi$, $l_0$ are the deformation energy, sample mass, twist angle and initial sample length, respectively. Theoretically, the torsional energy $\Delta E_t$ can be calculated from the twist angle $\varphi$ (in the unit of radian) through $\Delta E_t = GI_p\varphi^2/2l_0$ in the elastic regime. Here $G$ and $I_p$ represent the shear modulus and polar moment of inertia, respectively. According to Fig. 2a, the profile of the gravimetric torsional energy agrees well with the parabolic relationship up to a large twist rate, signifying that it follows Hook's law. Specifically, a high-torsional elastic limit around 0.57 rad nm$^{-1}$ is observed for the examined DNT bundle, corresponding to a torsional angle of about 8.6 rad or 493°. Here, torsional elastic limit is defined as the maximum twist rate before the occurrence of bond breaking, which is related to the maximum elastic energy that the structure can bear. As illustrated in Fig. 2b, there is no flattening during the twist process due to the ultrathin characteristic of the DNT, and the failure initiates from the fixed end of the structure. After the occurrence of bond breaking, obvious energy reduction is detected, which is associated with the fracture of individual DNT strands. For instance, the failure of the second DNT strand (Fig. 2b) is found to induce a strain energy release around 0.1 MJ kg$^{-1}$ (from B to C in Fig. 2a). We note that after each energy drop event, a hardening period usually follows, which is understandable, as the failure of DNT strand will release a certain amount of strain energy in the remaining intact DNT strands.

It is interesting to compare the twist deformation with the bundle structure made from CNT. In this regard, we took the most abundant (10,10) armchair nanotubes as an example. Here a same nanotube length of $\sim 17$ nm was adopted and loading region at each end is around 1 nm. The predictions agree with previous studies[11]. A slight torsional load would induce flattening of the CNT ($< 0.04$ rad nm$^{-1}$, point A′ in Fig. 2a). Due to the flattening (Fig. 2c), the gravimetric deformation energy accumulates quickly and the bundle structure is fractured at a very small twist rate, $\sim 0.16$ rad nm$^{-1}$, which is almost three

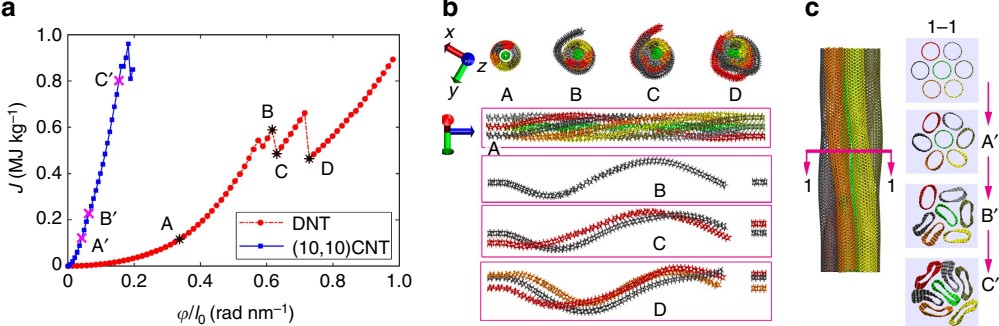

**Figure 2 | Comparisons of the twist deformation between DNT and CNT bundles.** (**a**) Gravimetric deformation energy density $J$ as a function of the twist rate $\varepsilon_{t0}=\varphi/l_0$. (**b**) Atomic configurations showing the structural change of DNT-based bundle at twist rate of 0.34, 0.62, 0.63 and 0.73 rad nm$^{-1}$, corresponding to points A to D in **a**. Top images are end-on views, and the corresponding side views are shown below (only the fractured DNT strands are visualized in B, C and D). (**c**) Atomic configurations illustrating the flattening of CNT-based bundle during twist deformation. Left image is the side view, and the right images show the transitions of the cross-section at different twist rates. Atoms are coloured based on the strand number in the fibre.

times smaller compared to that of the DNT bundle. These results indicate that the ultrathin DNT has very excellent torsional deformation capability compared to the abundant (10,10) CNT. It is worth noting that the applied loading rate might influence the estimated torsional elastic limit. In this regard, we repeated the test at another four loading rates with the twist period ranging from 4,000 to 8,000 ps on a single DNT. It is found that all torsional energy curves overlap with each other in the elastic regime and the elastic limit fluctuates with a small s.d. ($\sim$0.4%, see Supplementary Fig. 1). These results indicate that the results are independent of the applied twist rate.

We also compared the torsional properties between an individual DNT and (10,10) CNT. Similar as observed from the bundle structure, DNT has much higher-torsional elastic limit ($\sim$1.67 rad nm$^{-1}$) compared with that of the (10,10) CNT ($\sim$0.67 rad nm$^{-1}$). However, due to its ultrathin diameter, a very small torsional rigidity is estimated for the DNT (that is, $\sim$20 eV Å), which is nearly three orders smaller than that of the (10,10) CNT (that is, $\sim$18,689 eV Å). Despite that, by approximating the DNT as a solid shaft and the CNT as a circular hollow shaft (with a thickness of 3.4 Å), we find that the DNT has a comparable shear modulus as that of the (10,10) CNT. Specifically, the shear modulus for (10,10) CNT is about 423 GPa, which aligns well with the previous reported values (in the range of 300–547 GPa)[19]. In comparison, a shear modulus about 114 GPa is derived for the DNT. Revisiting the deformation energy curve of the seven strand DNT bundle, a torsional rigidity about 325 eV Å obtained (fitted with the twist rate within 0.2 rad nm$^{-1}$). In comparison, the (10,10) CNT bundle shows a torsional rigidity around $1.65 \times 10^4$ eV Å (fitted with the twist rate within 0.05 rad nm$^{-1}$ before excessive flatten), over 50 times larger than that of the DNT bundle (see detailed calculations in Supplementary Fig. 2). In addition, another three CNTs with smaller diameter were adopted to compare with the DNT, including (4,3) CNT, (0,8) CNT and (5,5) CNT. It is uniformly found that the CNT bundles have larger torsional rigidity but smaller torsional elastic limit than that of the DNT, whereas the gap between them narrows with decreasing CNT diameter (see Supplementary Fig. 3 and Supplementary Table 1). We should note that the SWD spacing in the studied DNT structure is at the minimum. The earlier work has shown that the stiffness of the DNT is controlled by the density of SWDs, that is, the less the SWD, the higher the stiffness[18]. Thus, the bundle constructed from DNTs with less SWDs have a higher-torsional stiffness, as shown in Supplementary Fig. 4.

It is also worth mentioning that the twist deformation of the DNT fibre is complex, which involves not only torsion but also

tension, compression and bending. Previous work on (10,10) CNT bundle (with strand number smaller than 19) suggested that torsion and tensile deformation are the dominant deformation mechanisms, and bending deformation is negligible[11]. Also, the deformation of DNT varies with its location, for example, the exterior DNT with larger distance to the twist axis will experience larger tensile and also bending deformation, but all DNTs have the same torsional angle. Considering the variations of the geometrical structures of DNT, understanding the contributions from different deformation components, as well as the deformation process in each constituent DNT requires a substantial work, such as that has been conducted in CNT bundles[11,20], which deserves further systematic studies.

**Interfacial load transfer.** One pivotal property for the fibre application is the interfacial load-transfer efficiency, which can be assessed through the pull-out test as frequently performed for nanocomposites[21,22]. Recall in Fig. 1f, the relaxed DNT bundle was further relaxed when periodic boundary conditions were removed and the right end of the core DNT and the six surrounding strands were fixed. Thereafter, a low constant velocity (0.02 Å ps$^{-1}$) was applied to the right end of the core DNT while keeping all six surrounding DNTs rigid. During the pull-out process, the 'transferred load' is taken as an index for the efficiency of the load transfer, which is calculated by summing all axial force acting on the core DNT. For comparison purpose, the pull-out test of a (10,10) CNT bundle was also carried out (with the same sample length).

Figure 3 compares the profile of the transferred load ($F$) between DNT and (10,10) CNT bundle as a function of the displacement ($d$) of the end of the core strand. Strikingly, although the $F$–$d$ curves experience relatively large fluctuations (as explained later), the transferred load maintains an almost constant average value for both DNT and CNT bundles until the complete pull-out of the core strand. We have also examined the DNT bundle with a smaller length ($\sim$10 nm), from which, a same result is obtained. Such results indicate that the transferred load is independent of the embedded length of the core strand (of either DNT or CNT bundle). This phenomenon is different from the common concept in bulk composite, but can be explained from the perspective of superlubricity as widely discussed for multi-walled carbon nanotubes[23,24], graphene/Au interface[25], and other rigid layered materials[26]. We will revisit this mechanism by exploring the pull-out process in CNT bundle in the following section. By averaging the recorded force from the displacement $d$ from 60 to 100 Å, the core DNT is found to experience a

transferred load of around 1.83 eV Å$^{-1}$, which is more than two times higher compared to that of the core (10,10) CNT (about 0.58 eV Å$^{-1}$). Here transferred load is only averaged for the displacement range from 60 to 100 Å in order to reduce the influence from the end of the bundle structure. Alternatively, following the conventional definition of shear strength $\tau = F/\pi DL$, DNT is estimated to have a shear strength of about 151 MPa, which is an order of magnitude larger than that of the CNT bundle ($\sim$12 MPa). Here, $L$ is approximated as 15 nm, and the diameter of DNT is estimated to be 4.12 Å according to the concept of linear atom density ($\delta$, in the unit of atoms per Å) based on the volume of the structure[14,27,28]. For (10,10) CNT, the external diameter is adopted, which is the summation of the CNT diameter (13.56 Å) and the graphite interlayer distance (3.35 Å). These estimations signify that the DNT exhibits much better load-transfer efficiency than that of (10,10) CNT in the bundle structure.

**Load transfer in a CNT bundle.** Before unveiling the load-transfer mechanisms in the DNT bundle structure, we first revisited the (10,10) CNT bundle structure. Ideally, there is only van der Waals (vdW) interaction between neighbouring CNTs. For perfect CNT shells, the intershell friction can be calculated from $F_{vdW} = -\pi D\gamma$, where $\gamma$ is the intershell cohesive energy density for a single shell (equals to 0.16 N m$^{-1}$)[23,29]. Adopting this equation, the friction force is around 0.43 eV Å$^{-1}$ for the CNT bundle, which is close to the results in Fig. 3. From the energy point of view, during the repetitive breaking and reforming of vdW forces (between core and surrounding CNTs) along with the pull-out process, two new surfaces are generated at the right end of the core strand and left end of the surrounding CNTs (pulling from left to right). As such, the corresponding surface energy change ($\Delta E_s$) equals to the work done by the transferred load, which is a linear function of the displacement increment ($\Delta x$), that is, $\Delta E_s = \pi D \Delta x (\gamma_{ss} + \gamma_{sc}) = F\Delta x$ (here, $\gamma_{ss}$ and $\gamma_{sc}$ represent the surface energy density of surrounding CNTs and core CNT, respectively). We should stress that the observed length independence of the transferred load in CNT bundle is consistent with the previous results from both MD simulations and *in situ* pull-out of multi-walled CNTs[24,30–33]. Theoretical studies have shown that there is a strong length effect on the load-transfer mechanisms during the pull-out of multi-walled CNT[34], that is the pull-out force will be independent of the overlapped length when the overlapped length is much smaller than a crossover length $L^*$ (which is over 200 nm). However, when the overlapped length is longer than the

crossover length, the interfacial friction is length-dependent, which is also predicted from the approximate mean-field-based model[35], and observed experimentally when the length of the CNT fibre is in the order of micrometre[36].

Besides this characteristic, the $F$–$d$ curve for the CNT bundle shows a relatively large fluctuation. From the enlarged view of the curve from the displacement of 60–70 Å (Fig. 4a), it is seen that the fluctuation has a very regular periodicity. Applying fast Fourier transformation to the $F$–$d$ curve (for 40 Å $<d<$ 130 Å), a single dominant frequency ($\sim$8.21 GHz) is obtained, which corresponds to a displacement of $\sim$2.44 Å. Correlating to the CNT structure, such displacement increment is coincident with the lattice constant, that is, $\sqrt{3}a$ ($a$ is the carbon bond length, $\sim$1.42 Å). In this regard, we looked at the interface commensurability of the bundle structure. Due to the same constituent CNTs, each contact surface between neighbouring CNTs is analogous to a bilayer graphene. As is known, the optimal stacking mode for bilayer graphene in terms of total energy is the parallel-displaced configuration (denoted as AB stacking, where half of the carbon atoms in top layer reside atop of the hexagonal centres of the bottom layer), and the worst stacking mode is the sandwich configuration (denoted as AA stacking, with the lattices of the two layers fully overlapped)[26,37]. From the bundle structure after energy minimization, we found that the core (10,10) CNT has four low-energy AB stacking interfaces and two high-energy AA stacking interfaces with the six surrounding CNTs (Fig. 4b). This interface is further seen when we examine an infinite triangular lattice of (10,10) CNT with six-fold symmetry, that is, each triangular CNT lattice has two AB and one AA interfaces (Supplementary Table 2). In other words, the core CNT has a kind of commensurate interface with the surrounding CNTs. With the pull-out along the zigzag direction, this commensurate interface changes repeatedly from an energy-minimum state to a high-energy state with a same period of $\sqrt{3}a$, that is, the transition of AA-SP$^1$-AA and AB-SP$^2$-AB in the interface. Here, SP$^1$ and SP$^2$ represent the two parallel-displaced configurations

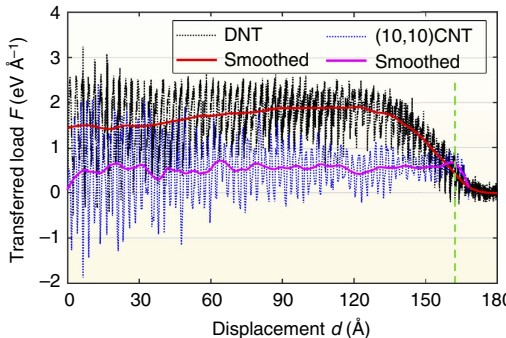

**Figure 3 | Comparisons of the pull-out test between DNT and CNT.** The transferred load as a function of the displacement of the end of the core strand for DNT and CNT bundles. Solid lines are the corresponding smoothed curves. Dashed green line indicates the complete pull-out of the core strand.

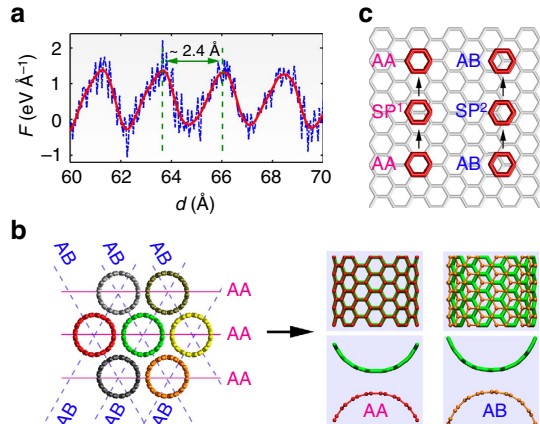

**Figure 4 | Interface load transfer in a CNT bundle.** (**a**) Enlarged $F$-$d$ curve of the CNT bundle, showing the highly regular periodic fluctuations of the transferred load. The red line is the smoothed curve. (**b**) Schematic view of the interface types between adjacent CNTs in the bundle structure (left of panel). Interfaces along the horizontal directions have an AA stacking configuration, and the rest along other inclined directions have an AB stacking configurations. The image on the right of the panel shows the side view (upper) and top view (lower) of two representative AA and AB stacking configurations at the interface. (**c**) Schematic view of the transition of AA-SP$^1$-AA and AB-SP$^2$-AB at the interface of CNT bundle during pull-out, with SP$^1$ and SP$^2$ representing the two parallel-displaced configurations between two equivalent AA or AB configurations, respectively.

between two equivalent AA or AB configurations, respectively (Fig. 4c). This periodic transition thus induces a periodic fluctuation to the transferred load, and endows a wave-shape profile. It is worth noting that the radial breathing mode of CNT might also influence the interface transferred load[38–40]. However, since the radial breathing mode is interrelated with the interface vdW interaction[39], a simple isolation of its influence on the transferred load is unavailable, which deserves further investigation (see more discussion in Supplementary Fig. 5).

**Stick–slip motion in DNT bundle.** Although the $F$–$d$ curve for the DNT bundle also exhibits a periodic fluctuation pattern, its profile is in a saw-toothed shape rather than the regular sine wave shape seen in the CNT bundle. As shown in Fig. 5a, each sawtooth has a width around 2.5 Å, which is about half of the unit length of the DNT. This interesting sawtooth feature implies that DNT bundle has totally different load-transfer mechanisms to that of the CNT bundle. Due to its ultrathin structure and a diamond-like tetrahedral motif, the examined DNT does not have radial breathing mode and aromatic surfaces (or $\pi$–$\pi^\star$ intertubule interactions). Rather, the hydrogenated surface of the DNT endows the bundle structure with an interface similar as that between hydrogenated diamond-like carbon films, which is supposed to have an extremely low friction coefficient, or be superlubricity[41]. As is widely accepted, hydrogen will passivate the unoccupied or free $\sigma$-bonds of the carbon atom, which will make the carbon atoms become chemically inert and result in very little adhesive interactions (that is, low friction) during sliding[42]. Contrary to this, a large transferred load is detected for the DNT bundle, which can be explained from two aspects. First, during the pull-out process, the DNT behaves essentially like a straight cylinder. Analogous to the CNT, the work done by the transferred load equals to the surface energy of the newly created

surfaces during the pull-out. That is, the transferred load is independent of the overlapped length of the core DNT. Second, the high density of the SWDs endows the DNT with a zigzag morphology, that is, a zigzag vdW surface[22], which introduces a stick–slip motion to the core DNT. Such stick–slip motion induces a high kinetic or dynamic friction during pull-out.

From the insights of the atomic configurations, the core DNT has two stable minima configurations including the overlapped status (OS, see inset A-OS in Fig. 5b), and the parallel-shifted status (PS, inset C-PS in Fig. 5b). For the overlapped status, atoms of the core DNT have exactly the same coordinates along the length direction with that of the surrounding DNTs (analogous to the AA stacking in bilayer graphene). For the parallel-shifted status, atoms of the core DNT shifted about half of its unit length ($\sim 2.5$ Å) along the axial direction of the OS (analogous to the AB stacking). The transitions of the core DNT between different statuses are illustrated in Fig. 5b by taking the energy-minimum status OS as beginning (inset A-OS in Fig. 5b). As is seen, only a minor shift of the lattice (close to the right end of the surrounding DNTs) is appeared when the core DNT changes from OS to strained status, as reflected by the small distance between the dashed and solid blue lines in inset B-SS in Fig. 5b. In comparison, from strained status (B-SS) to parallel-shifted status (C-PS), the lattice experiences a sudden large shift in the pulling direction (see dashed and solid blue lines in inset C-PS in Fig. 5b). Afterwards, the core DNT changes again from the energy-minimum status (C-PS) to a strained status (D-SS), and thus a minor shift of the lattice is observed. We should re-emphasize that the displacement in Figs 3a and 5a is recorded from the loading end of the core DNT, which is not the movement of the mass centre of the whole core DNT.

Evidently, during the pull-out, a significant force climb will occur when the DNT is away from its stable status, and the core DNT will be stretched into a strained status (SS) rather than

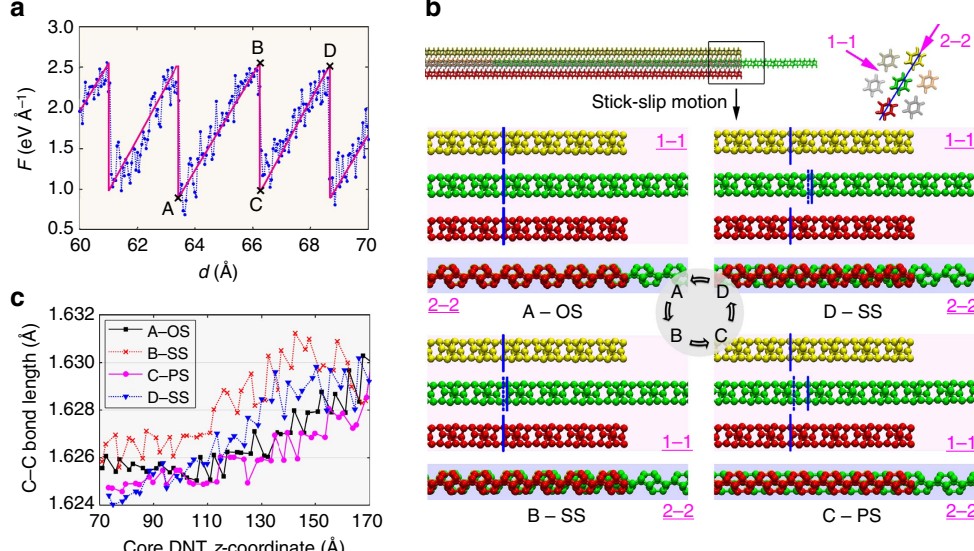

**Figure 5 | Interface load transfer in a DNT bundle.** (**a**) Enlarged $F$–$d$ curve of the DNT bundle, showing the highly periodic fluctuations of the transferred load and also a saw-toothed feature. The magenta line is the schematic representation of the sawtooth curve. (**b**) Atomic configurations illustrating the stick–slip motion of the core DNT during pull-out. The top image is of the whole DNT bundle structure and upper right image shows the cross-sectional view, which illustrates the two viewing directions, that is, 1–1 and 2–2. Insets A-OS, B-SS, C-PS and D-SS are the corresponding atomic configurations at the displacements corresponding to A, B, C and D in **a**, in which only the three strands along the 2–2 directions are visualized for clarity. Here only C atoms near the right end of the surrounding DNTs are visualized for easier identification of the stick–slip phenomenon and they are coloured based on the strand number in the fibre. The blue lines in the 1–1 views of insets A-OS, B-SS, C-PS and D-SS illustrate the actual displacement/shift of the whole DNT from A–B, B–C and C–D, respectively. Here, the middle dashed and solid blue lines represent the location of the tracked atoms in previous and current status, which clearly show the shift of the core DNT between two statuses. (**c**) Comparisons of the C–C bond length of the core DNT longer than 1.61 Å at the four corresponding states in **b** in the overlapped region from 70 to 170 Å ($z$-coordinate).

moving along the pulling direction (regions A–B and C–D in Fig. 5a). Once the pulling force is large enough to cross the energy barrier, the DNT will suddenly change from configuration SS to an adjacent energy-minimum status, reflected by a sudden sharp reduction of the transferred load (region B–C in Fig. 5a). The strained status is also readily seen from the bond length distribution of the core DNT at the four representative configurations (that is, A-OS, B-SS, C-PS, and D-SS). In the calculation of bond-length distribution, the overlapped section of the core DNT from 70 to 170 Å (z-coordinate) is considered. As compared in Fig. 5c, the bond lengths of the core DNT for the two strained status (B-SS and D-SS in Fig. 5a) are significantly longer than those in the cases of A-OS and C-PS. Here, only the stretched C–C bond longer than 1.61 Å is considered, a complete comparison of the C–C bond length distribution of the core DNT can be found in Supplementary Fig. 6. Overall, the repetition of the transition of OS-SS-PS-SS (or A-B-C-D in Fig. 5a) during the pull-out introduces a strong mechanical interlocking effect, that is, a stick–slip motion, which therefore results in a saw-toothed transferred load profile. In other words, although the DNT bundle has a fully hydrogenated diamond-like film interface, its highly commensurate interfaces induce a much higher load-transfer efficiency compared to that of the (10,10) CNT bundle. To note that the current study has focused on a typical seven strand bundle. According to the results on CNT fibres, there exist obvious scale effects resulted from their diameter and length (that is, the strand number and strand length). For instance, in CNT bundles with larger diameter, the twist-induced tension, bending and compressive deformation play a more important role compared with that in the smaller bundles[11,20,43]. On the other hand, as aforementioned, the interface force/friction during the pull-out of the multi-walled CNT will rely on the overlapped length when its length is much larger than a crossover length[34]. Thus, a further thorough investigation is still anticipated to unveil the scale effect on the mechanical properties of DNT bundles.

To further verify the observation that DNT bundle possesses much better interface load-transfer efficiency than CNT bundle, we investigated the sliding behaviour of another three smaller CNT bundles, including (4,3) CNT, (0,8) CNT and (5,5) CNT. It is found that all CNT bundles exhibit a constant averaged transferred load during sliding, which is much smaller compared with that of the DNT, ranging from 0.31 to 0.34 eV Å$^{-1}$ (see Fig. 6a and Supplementary Fig. 7). Following the above calculation, the interface shear stress fluctuates around 11.5 MPa for the CNT bundles, which decreases gradually with the increasing diameter of the constituent CNT. Comparing with the interface shear strength or transferred load, these results uniformly signify that DNT bundles has superior interface load-transfer efficiency than that of the CNT bundles.

It should be mentioned that there are three types of nanothreads including achiral, stiff chiral and soft chiral (classified according to their geometrical characteristics)[14], and the current work has focused on the DNT with Stone–Wales transformation defects. Above discussions have clearly shown that the effective interface load-transfer efficiency for the DNT bundle originates from the strong mechanical interlocking effect, which is resulted from the zigzag morphology or surface irregularities of the constituent DNTs. Therefore, it is expected that nanothreads with highly irregular surface will possesses a high interface load-transfer efficiency (a complete comparison of the geometrical structures of the DNTs can be found from the work by Xu et al.[14]). In this regard, we have selected another two representative DNT bundles, one is the straight DNT—with no SWD and has a smooth surface, and the other one belongs to the soft chiral group, which has a stable spiral structure—with a highly irregular surface (Fig. 6b). Following the same simulation settings, we find that the straight DNT has a very small transferred load around 0.25 eV Å$^{-1}$, whereas the spiral nanothread exhibits a high maximum transferred load ($\sim 3.3$ eV Å$^{-1}$) at the early stage of the sliding, which leads to the bond breaking of the structure. This result indicates that the interfacial load can be loaded up to the elastic limit of the soft chiral nanothread. By assuming that the core DNT is under a pure tensile deformation, such maximum transferred load equals to a tensile stress of $\sim 35$ GPa. Overall, these results have well verified our assumption that smooth surface nanothreads will possess a low interface load-transfer efficiency, while the nanothreads with highly irregular surface will have a very high load-transfer efficiency. The details are given in Supplementary Fig. 8.

## Discussion

Compared with the most abundant (10,10) CNT, DNT does not flatten during twist deformation. Specifically, the DNT bundle structure is found to have a high-torsional elastic limit around 0.57 rad nm$^{-1}$, which is almost four times that obtained for the (10,10) CNT bundle, indicating an excellent torsional deformation capability. More importantly, according to the pull-out tests, the DNT bundle exhibits an interface transferred load fluctuating around 1.83 eV Å$^{-1}$, which is more than twice of that of the core (10,10) CNT (about 0.58 eV Å$^{-1}$). Adopting the conventional definition of the shear strength (that is, force over contact area), the DNT bundle has an interface shear strength about 151 MPa. This is an order of magnitude larger than that of the CNT bundle ($\sim 12$ MPa). These estimations signify that the DNT exhibits much better load-transfer efficiency than (10,10) CNT in the bundle structure. It is found that the same constituent CNTs

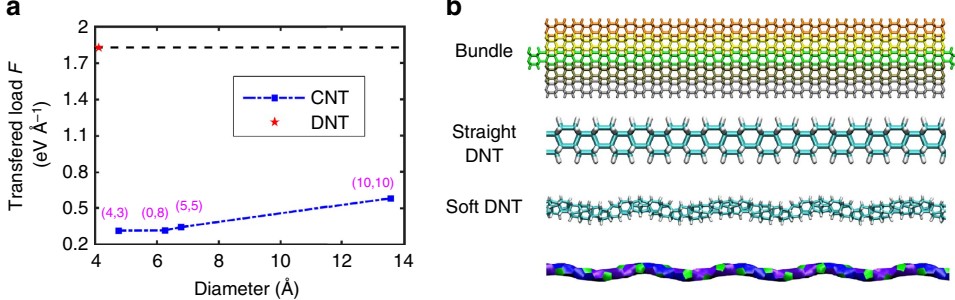

**Figure 6 | Load transfer in different DNT and CNT bundles.** (**a**) Comparisons of the transferred load. (**b**) DNT bundles constructed from the smooth DNT (without SWD), and a spiral soft DNT (bottom image clearly shows the spiral feature of the soft DNT). To avoid the spiral-end influence, the core strand is longer than the surrounding DNTs (one-unit length in each end). A shorter bundle has been considered as the transferred load is independent of the overlapped length, with the surrounding strands approximate to 10 nm. The sliding load was imposed on the extrude end of the core DNT.

endow the CNT bundle with a kind of commensurate interface, and thus induce a wave-shaped load transfer profile. Unexpectedly, although the fully hydrogenated surface is expected to induce an extremely low friction coefficient, the excellent load-transfer efficiency in the DNT bundle structure is attributed to the zigzag morphology of the DNT (as resulted from the SW transformation defects). During the pull-out, the zigzag surface induces a strong mechanical interlocking effect at the interface through the stick–slip motion, which greatly enhanced the load-transfer efficiency.

In summary, our *in silico* results suggest that DNT is an ideal candidate for fibre applications. It not only possesses excellent torsional deformation capability, but also has excellent load-transfer efficiency. More importantly, the $sp^3$ structure should make it easy to build inter-thread cross-links, which will provide stronger interface load-transfer efficiency. Further, given that the studied DNT is a representative structure of the diamond nanothread family, a large variety of DNT fibres could potentially be made with highly tunable mechanical properties.

## Methods

**Empirical potentials.** For all simulations, the widely used adaptive intermolecular reactive empirical bond order (AIREBO) potential was employed to describe the C–C and C–H atomic interactions[44,45]. This potential includes short-range interactions and long range van der Waals interactions, which has been shown to well represent the binding energy and elastic properties of carbon materials. It contains dihedral terms that describe the torsional interactions, and thus enable to accurately reproduce the elastic properties of carbon systems, such as the compression of CNT bundles[46], torsion of CNT[47,48], tension-twisting deformation of CNTs[49]. Moreover, it adopts Lennard–Jones term to describe the van der Waals interactions, which has been reported to reasonably capture the vdW interactions in multi-layer graphene[50], multi-wall carbon nanotubes[51], carbon nanotube bundles[46] and hybrid carbon structure[52]. The cut-off distance of the AIREBO potential was chosen as 2.0 Å (refs 53–58). To note that the Lennard–Jones term is reported to be deficient in describing the $sp^2$ interlayer interactions with changes in the interlayer registry[30], which however will not affect our results as this work focused on DNTs and they are $sp^3$ carbon structures.

**Molecular dynamics simulations.** The DNT structures were first optimized by the conjugate gradient minimization method and then equilibrated using Nosé–Hoover thermostat[59,60] for 1 ns. Periodic boundary conditions were applied along the length direction during the relaxation process. To limit the influence from the thermal fluctuations, a low temperature of 1 K was adopted for all simulations. After relaxation, a pair of constant torsional load (that is, $2\pi/12,000$ rad ps$^{-1}$) was applied to twist the model (equal to a period of 6,000 ps, along its axis). All twisting simulations were carried out by switching to non-periodic boundary conditions, and one end of the sample was fully fixed with the other end free in longitudinal direction. A small time step of 0.5 fs was used for all calculations with all MD simulations being performed under the software package LAMMPS[61].

**Data availability.** The data that support the findings of this study are available from the corresponding authors on reasonable request.

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

## Acknowledgements

Supports from the ARC Discovery Project (DP150100828) and the High Performance Computer resources provided by the Queensland University of Technology are gratefully acknowledged. H.Z. is grateful to Dr Tao Liang from Pennsylvania State University for the discussion on the AIREBO potentials.

## Author contributions

H.Z. carried out the simulation, H.Z., G.Z., V.B.C.T. and Y.G. conducted the analysis and discussion.

## Additional information

**Competing interests:** The authors declare no competing financial interests.

**Publisher's note**: 

