## [Peer Review File · Nature Communications]

Reviewers' comments:

Reviewer #1 (Remarks to the Author):

The manuscript by Zhan *et al.* is a computational investigation of built-up carbon nanofibers. Specifically, it compares fibers composed with carbon nanotubes (CNTs) to fibers composed with the newly discovered/synthesized diamond nanothreads (DNTs). The primary conclusions are that there is an increased torsional capacity combined with interfacial stress transfer, resulting in a more robust fiber. The modeling is robust, and the conclusions are warranted by the results. The emerging DNT is an interesting platform, and will be of general interest, particularly for those working with CNT-based fiberlike systems. However, there are a few key issues that should be addressed. In particular:

1. The torsional stiffness of a *single* DNT should be quantified and compared to the torsional stiffness of a CNT. I would assume the sporadic SW defects in the DNT would actually result in rather compliant twisting. It would be interesting to compare the polar rigidities (e.g., GJ or GI_p) values and effective shear modulus, *G*.
2. Related to the above, if the torsional stiffness of the DNT is (relatively) weak, then the twisting deformation of the nanofiber would be dominated by the bending resistance of the DNTs, i.e., the curvature of the threads (which is somewhat suggested by the deformed shapes in **Figure 2b**).
3. The parabolic nature of energy density versus twist rate (**Figure 2a**) of the DNT fiber suggests a linearelastic torsional stiffness. Please fit the data and calculate an equivalent GJ for comparison with the CNT bundle.
4. What effect would the SWD distribution have on the torsional behavior of the bundle? Observing **Figure 1**, it appears the SWD spacing is rather short (at the minimum?).
5. Also, related to the DNT structure, are the hydrogens explicitly modeled?
6. An interesting aspect would be the energy per DNT/CNT in the bundle. E.g., do the exterior threads/tubes undergo less deformation than the interior threads? By how much?
7. Related to the above, is there a scale effect? The "bundles" here are the minimum required to create a packed structure (e.g., a core thread/tube with six surrounding threads/tubes). Would larger bundles exploit the same deformation mechanisms observed here? This should at least be discussed.
8. The (10,10) tubes used here are rather large. Would the change in behavior be so dramatic if smaller CNTs were used, such as (5,5) tubes? In particular, the smaller the tube diameter, the less susceptible the tubes are to buckling. This would be critical to make an apt comparison, considering the size difference between the DNTs and (10,10) tubes.
9. Due to the structure of DNTs with sp³ structure and hydrogens (unlike carbon-only sp² structure of CNTs), there is potential for inter-thread cross-links. Again – while a single study cannot explore every aspect - this should be mentioned as a potential advantage.
10. Finally, the interfacial load transfer is promising. One issue with CNTs is that for a longer thread, the nanotubes are not continuous. Thus, the tubes must be sufficient length to maximize the transfer of stress. DNTs have higher interfacial shear strength, such that the necessary critical length will be shorter. Can this length be calculated/shown?

Reviewer #2 (Remarks to the Author):

This is an interesting and timely paper on mechanical properties of nanothread bundles. I do have some important concerns however that need to be addressed.

First, the precise atomic structure of nanothreads is not yet known - many possibilities have been

enumerated as indicated by the references in the introduction. The load transfer between threads, for example, is likely to be sensitive to the precise structure of the constituent threads. Thus this issue should be explored more thoroughly, or else it is not known how widely the results obtained here apply to real threads.

Second, it is unclear how well the inter atomic potential chosen can handle variations in the inter-tube interaction with variations in registry. There are, for example, registry-dependent potentials that indicate that a simple LJ treatment of this interaction is much too smooth (and AIREBO uses a not so different form for this component of the interaction). Furthermore, the accuracy of this potential under transverse compression of the bundle (such as would be seen under torsion) needs to be addressed.

Finally, it is possible for some of the high-symmetry thread structures to have a radial breathing mode, contrary to the assertion made here.

Reviewer #3 (Remarks to the Author):

While the analysis is solid and the manuscript is well written, the lack of experimental validation undermines the potential impact of this work. In fact, it wasn't clear when I first reviewed the abstract that the work is entirely theoretical in nature (especially the second sentence reads "In this work, we find that the recently synthesized ultrathin diamond nanothread not only possesses excellent torsional deformation,....").

I think the work is more suitable for publications in other journals (e.g., Carbon, Nanoscale, and NanoLetters). The conclusion that "the load transfer is independent of overlapped length" is somewhat surprising and (at least for the CNT case) contradicts both simulation results by Yakobson et al. and experimental work as reviewed in Behabtu, Natnael, Micah J. Green, and Matteo Pasquali. "Carbon nanotube-based neat fibers." Nano Today 3.5 (2008): 24-34.

Response to Reviewer 1

Review Comments: *The manuscript by Zhan et al. is a computational investigation of built-up carbon nanofibers. Specifically, it compares fibers composed with carbon nanotubes (CNTs) to fibers composed with the newly discovered/synthesized diamond nanothreads (DNTs). The primary conclusions are that there is an increased torsional capacity combined with interfacial stress transfer, resulting in a more robust fiber. The modeling is robust, and the conclusions are warranted by the results. The emerging DNT is an interesting platform, and will be of general interest, particularly for those working with CNT-based fiber like systems.*

Author reply: We are grateful to the reviewer for commenting our work as interesting and robust study. We also thank the reviewer for her/his constructive comments and suggestions. Below we give our detailed response in a point-to-point manner. The manuscript has been revised accordingly (see the list of changes.)

Specific Comments:

Comment 1: The torsional stiffness of a single DNT should be quantified and compared to the torsional stiffness of a CNT. I would assume the sporadic SW defects in the DNT would actually result in rather compliant twisting. It would be interesting to compare the polar rigidities (e.g., GJ or GI_p) values and effective shear modulus, G.

Author reply: Considering that the torsional rigidity GI_p is a product of the shear modulus G and the polar moment of inertia I_p , the DNT is supposed to have a much smaller torsional rigidity compared with that of the CNT due to its ultrathin diameter. In the revision, we have added the comparisons of the estimated torsional rigidity and shear modulus between the studied DNT and CNT. It is found that although DNT has much smaller torsional rigidity than that of the (10,10)CNT, their shear modulus is comparable.

In the revision, following discussions have been added in the main manuscript. And one new figure (Figure S2) and new section S2 have been added in the Supporting Information:

On Page 7: “We also compared the torsional properties between an individual DNT and (10,10) CNT (see **Supporting Information, S2**). Similar as observed from the bundle structure, DNT has much higher torsional elastic limit (~1.67 rad/nm) compared with that of the (10,10) CNT (~0.67 rad/nm). However, due to its ultrathin diameter, a very small torsional rigidity is estimated for DNT (i.e., ~20 eV·Å), which is nearly three orders smaller than that of the (10,10) CNT (i.e., ~18689 eV·Å). Despite that, by approximating the DNT as a solid shaft and the CNT as a circular hollow shaft (with a thickness of 3.4 Å), we find that the DNT has a comparable shear modulus as that of the (10,10) CNT. Specifically, the shear modulus for (10,10) CNT is about 423 GPa, which aligns well with the previous reported values (in the range of 300 to 547 GPa).¹⁹ In comparison, a shear modulus about 114 GPa is derived for the DNT.”

Comment 2: *Related to the above, if the torsional stiffness of the DNT is (relatively) weak, then the twisting deformation of the nanofiber would be dominated by the bending resistance of the DNTs, i.e., the curvature of the threads (which is somewhat suggested by the deformed shapes in Figure 2b).*

Author reply: We agree with the reviewer that the bending deformation might play an important role during the twist deformation of the DNT bundle. This issue has been investigated for CNT fibers. Theoretically, along with the twisting of the CNT fiber, each constituent strand deforms into a coil of radius ρ , involving stretching, compression, bending and twisting. Particularly, during the

twist, the bending energy for each CNT strand i can be calculated from $\Delta E_{b,i} / l_0 = \beta(r_0 / R_i)^2$ (r_0 is the radius of the CNT), where the bending radius of each coil can be approximated as $R_i = \rho_i [1 + (l_0 / \varphi \rho_i)^2]$ (Teich *et al*, *Phys Rev Lett*, **2012**, 108, 235501). For the CNT bundle (with strand number <19), studies have shown that the torsion and tension are the two dominated deformation process during twisting, and the bending deformation is ignorable (Teich *et al*, *Phys Rev Lett*, **2012**, 109, 255501). Previous works from other group (Roman *et al*, *Nano Lett*, **2015**, 15, 1585-1590.) and ours (Zhan *et al*, *Carbon*, **2016**, 107, 304-309) have shown that the DNT has a bending stiffness approaching 1480 kcal/mol·Å, about two orders lower than that of CNT (Roman *et al*, *Nano Lett*, **2015**, 15, 1585-1590). Additionally, we note that insets B-D in Figure 2b only present the fractured DNT strands, which retains a curved shape due to the van der Waals interactions between the fractured and un-fractured DNT strands. Overall, the twist deformation of DNT bundle/fiber is a combination of twist, tension, compression and bending. However, a thorough investigation is still anticipated, which will be our following work.

In the revision, following discussions have been added to make it more comprehensive:

On Page 8: “It is also worth mentioning that the twist deformation of the DNT fiber is complex, which involves not only torsion but also tension, compression and bending. Previous work on (10,10) CNT bundle (with strand number smaller than 19) suggested that torsion and tensile deformation are the dominant deformation mechanisms, and bending deformation is ignorable.¹¹ Also, the deformation of DNT varies with its location, e.g., the exterior DNT with larger distance to the twist axis will experience larger tensile and also bending deformation, but all DNTs have the same torsional angle. Considering the variations of the geometrical structures of DNT, understanding the contributions from different deformation components, as well as the deformation process in each constituent DNT requires a substantial work, such as that has been conducted in CNT bundles,^{11,20} which deserves further systematic studies.”

Comment 3: *The parabolic nature of energy density versus twist rate (Figure 2a) of the DNT fiber suggests a linear elastic torsional stiffness. Please fit the data and calculate an equivalent GJ for comparison with the CNT bundle.*

Author reply: As suggested, the equivalent torsional rigidity GI_p has been estimated from the energy versus twist rate curve. Considering the flatten phenomena, only the energy curve with twist rate smaller than 0.05 rad/nm has been adopted during the fitting for the CNT bundle. Whereas, a larger twist rate ranging from 0 to 0.2 rad/nm has been used for the DNT bundle. The (10,10) CNT bundle shows a torsional rigidity around 1.65×10^4 eV·Å, over 50 times larger than that of the DNT bundle. In comparison, for the individual CNT and DNT, the torsional rigidity of (10, 10) CNT is about three orders of magnitude higher than that of the single DNT.

In the revision, following discussions have been added.

On Page 7-8: “Revisiting the deformation energy curve for the seven strand DNT bundle, a torsional rigidity about 325 eV·Å is obtained (fitted with the twist rate within 0.2 rad/nm). In comparison, the (10,10) CNT bundle shows a torsional rigidity around 1.65×10^4 eV·Å (fitted with the twist rate within 0.05 rad/nm before excessive flatten), over 50 times larger than that of the DNT bundle.”

Comment 4: *What effect would the SWD distribution have on the torsional behavior of the bundle? Observing Figure 1, it appears the SWD spacing is rather short (at the minimum?).*

Author reply: As discussed earlier, the torsional behavior of the bundle involves twist, tensile, bending, and compressive deformation. This work has adopted the model with the SWD spacing at the minimum. Earlier works from other group (Roman *et al*, *Nano Lett*, **2015**, 15, 1585-1590) and ours (Zhan *et al*, *Nanoscale* **2016**, 8 (21), 11177-11184.) have shown that adding SWDs will

introduce ductile characteristic to DNT, i.e., DNT with less SWD is stiffer. Thus, it is expected that the bundle constructed from DNTs with less SWDs will have higher torsional rigidity.

In the revision, we have done additional calculations, added one Figure (Figure S4), one section (see new section S4 in Supporting Information), and added following discussions in the main manuscript to justify this point.

On Page 8: “We should note that the SWD spacing in the studied DNT structure is at the minimum. Earlier work has shown that the stiffness of the SWD is controlled by the density of SWDs, that is, the less the SWD, the higher the stiffness.¹⁸ Thus, the bundle constructed from DNTs with less SWDs have a higher torsional stiffness, as shown in Figure S4 in **Supporting Information**.”

Comment 5: *Also, related to the DNT structure, are the hydrogens explicitly modeled?*

Author reply: Yes. In the simulation, the hydrogens are explicitly modeled. We note that only C atoms are visualized in Figure 5 for easier identification of the stick-slip phenomenon. In the revision, we have added one sentence in the caption of Figure 5.

On Page 16: “Here only C atoms near the right end of the surrounding DNTs are visualized for easier identification of the stick-slip phenomenon and they are colored based on the strand number in the fiber.”

Comment 6: *An interesting aspect would be the energy per DNT/CNT in the bundle. E.g., do the exterior threads/tubes undergo less deformation than the interior threads? By how much?*

Author reply: It is sure that the strand undergoes different deformation depending on its locations.

For example, the tensile strain rate in each strand can be estimated from $\epsilon_{s,i} = \sqrt{1 + (\rho_i \phi / l_0)^2} - 1$ (Teich *et al*, *Phys Rev Lett*, **2012**, 109, 255501). Apparently, the exterior strand with larger radius ρ (with respect to the twist axis) will experience larger tensile and also bending deformation, but all strands have the same torsional angle. Similar works have already been conducted on CNT bundles, e.g., Teich *et al* (*Phys Rev Lett*, **2012**, 109, 255501) illustrated the contribution of torsion, tension, compression and bending during bending from the energy storage perspective; Zhao *et al* (*J Mech Phys Solids*, **2014**, 71, 64-83) developed a theoretical model to describe the deformation in the bundle structures based on the deformation of each strand. However, as also mentioned in the reply to Comment #2, understanding the deformation in each constituent DNT requires a substantial work to analysis the twist deformation process of the DNT bundle, which will be the focus of our following work.

In the revision, following discussions are added in combination with the reply to Comment #2:

On Page 8: “It is also worth mentioning that the twist deformation of the DNT fiber is complex, which involves not only torsion but also tension, compression and bending. Previous work on (10,10) CNT bundle (with strand number smaller than 19) suggested that torsion and tensile deformation are the dominant deformation mechanisms, and bending deformation is ignorable.¹¹ Also, the deformation of DNT varies with its location, e.g., the exterior DNT with larger distance to the twist axis will experience larger tensile and also bending deformation, but all DNTs have the same torsional angle. Considering the variations of the geometrical structures of DNT, understanding the contributions from different deformation components, as well as the deformation process in each constituent DNT requires a substantial work, such as that has been conducted in CNT bundles,^{11,20} which deserves further systematic studies.”

Comment 7: *Related to the above, is there a scale effect? The “bundles” here are the minimum required to create a packed structure (e.g., a core thread/tube with six surrounding threads/tubes). Would larger bundles exploit the same deformation mechanisms observed here? This should at least be discussed.*

Author reply: We thank the reviewer for this valuable suggestion. Yes, there exists scale effect regarding the mechanical properties of the bundles, arising from their diameter and length (i.e., the strand number and strand length). Several studies have discussed the deformation of CNT bundles with more number of strands (Zhao *et al*, *J Mech Phys Solids*, **2014**, 71, 64-83; Xiang *et al*, *Int J Solids Struct*, **2015**, 58, 233-246). In principle, the contribution from different deformation components (as accompanied with twist deformation) will vary with the size of the bundle. In addition, the strand length will affect the load transfer mechanisms between DNTs (see reply to Comment #10).

In the revision, following discussion has been added.

On Page 15: “To note that the current study has focused on a typical seven strand bundle. According to the results on CNT fibers, there exist obvious scale effects resulted from their diameter and length (i.e., the strand number and strand length). For instance, in CNT bundles with larger diameter, the twist-induced tension, bending and compressive deformation play a more important role compared with that in the smaller bundles.^{11,20,41} On the other hand, as aforementioned the interface force/friction during the pull-out of the multi-walled CNT will rely on the overlapped length when its length is much longer than a crossover length.³⁴ Thus, a further thorough investigation is still anticipated to unveil the scale effect on the mechanical properties of DNT bundles.”

Comment 8: *The (10,10) tubes used here are rather large. Would the change in behavior be so dramatic if smaller CNTs were used, such as (5,5) tubes? In particular, the smaller the tube diameter, the less susceptible the tubes are to buckling. This would be critical to make an apt comparison, considering the size difference between the DNTs and (10,10) tubes.*

Author reply: We thank the reviewer for this valuable suggestion. The (10,10) CNT was selected in this study as it is the most abundant CNTs in experiments. To make the comparison more comprehensive, we have added another three smaller tubes, including (4,3), (0,8), and (5,5) CNTs. Specifically, (4,3) CNT is one of the thinnest CNTs being reported experimentally (Guan *et al*, *Nano Lett*, **2008**, 8, 459-462). Both twist and sliding simulations have been carried for the three CNT bundles, from which we obtain consistent results compared with the (10,10) CNT bundle. In brief, for the twist deformation, CNT bundles possess much higher torsional rigidity as resulted from better mechanical properties and larger polar moment of inertia. Whereas, all examined CNTs shows smaller torsional elastic limit compared with that of the DNT. From the sliding simulations, as expected, all CNT bundles exhibit a low transferred load, which increases slightly with the diameter of the constituent CNT. Adopting the conventional definition of shear strength $\tau = F / \pi DL$, the CNT bundle is showing a decreasing interface shear strength with the diameter (which is around one order smaller compared with that of the DNT).

In the revision, we have added calculations of another three CNTs. We have added one figure (Figure 6), two paragraphs in the main manuscript, and two figures (Figures S3 & S8) and two sections (S3 & S8) in the Supporting Information. Following discussions have been added in the manuscript.

On Page 8: “In addition, another three CNTs with smaller diameter were adopted to compare with the DNT, including (4,3) CNT, (0,8) CNT, and (5,5) CNT (see **Supporting Information, S3**). It is uniformly found that the CNT bundles have larger torsional rigidity but smaller torsional elastic limit than that of the DNT. Whereas, the gap between them narrows with decreasing CNT diameter.”

On Page 17: “To further verify the observation that DNT bundle possesses much better interface load transfer efficiency than CNT bundle, we investigated the sliding behavior of another three smaller CNT bundles, including (4,3), (0,8), and (5,5) CNT. From Figure S8 in Supporting Information, it is clearly that all CNT bundles exhibit a constant averaged transferred load during sliding, which is much smaller compared with that of the DNT, ranging from 0.31 to 0.34 eV/Å (see **Figure 6a**). Following the above

calculation, the interface shear stress fluctuates around 11.5 MPa for the CNT bundles, which decreases gradually with the increasing diameter of the constituent CNT (see **Supporting Information S8** for more details). Comparing with the interface shear strength or transferred load, these results uniformly signify that DNT bundles has superior interface load transfer efficiency than that of the CNT bundles.

Figure 6 (a) Comparisons of the transferred load between DNT bundle and CNT bundles. (b) DNT bundles constructed from the smooth DNT (without SWD), and a spiral soft DNT (bottom image clearly shows the spiral feature of the soft DNT). To avoid the spiral end influence, the core strand is longer than the surrounding DNTs (one-unit length in each end). A shorter bundle has been considered as the transferred load is independent of the overlapped length, with the surrounding strands approximate to 10 nm. The sliding load was imposed on the extrude end of the core DNT.”

Comment 9: Due to the structure of DNTs with sp^3 structure and hydrogens (unlike carbon-only sp^2 structure of CNTs), there is potential for inter-thread cross-links. Again – while a single study cannot explore every aspect - this should be mentioned as a potential advantage.

Author reply: We thank the reviewer for this valuable suggestion. In the revision, following discussion has been added.

On Page 19: “More importantly, the sp^3 structure will make it easy to build inter-thread cross-links, which will provide stronger interface load transfer efficiency.”

Comment 10: Finally, the interfacial load transfer is promising. One issue with CNTs is that for a longer thread, the nanotubes are not continuous. Thus, the tubes must be sufficient length to maximize the transfer of stress. DNTs have higher interfacial shear strength, such that the necessary critical length will be shorter. Can this length be calculated/shown?

Author reply: We thank the reviewer for this valuable suggestion. Earlier works on CNT fiber from the approximate mean-field based model (Yakobson *et al*, *Carbon*, **2000**, 38, 1675-1680) has shown that the critical length (L_m) of the CNT should be in the order of 10 μm to approach the ideal value of strength. This critical length can be roughly estimated based on the length-dependent lateral friction and the yield strength. According to previous theoretical analysis (Huhtala *et al*, *Phys. Rev. B* **2004**, 70 (4), 045404), there exists a crossover length L^* (over 200 nm for CNT), under which the interlayer friction in multi-walled CNT is independent of the overlapped length. To estimate the critical length L_m , a CNT longer than this crossover length is necessary. In our current MD simulations, the transferred load is independent on the overlapped length of DNTs. Such results are also observed from our additional simulation with a shorter length about 10 nm. In order to calculate the critical length L_m for DNT, much longer DNT model (hundreds of nm) should be considered in order to unveil the length effect on the interface load transfer mechanisms. In the revision, we added the following discussion to further justify the scale effect in combination with the reply to Comment #7.

Page 9: “We have also examined the DNT bundle with a smaller length (~ 10 nm), from which, a same result is obtained.”

Page 15: “To note that the current study has focused on a typical seven strand bundle. According to the results on CNT fibers, there exist obvious scale effects resulted from their diameter and length (i.e., the strand number and strand length). For instance, in CNT bundles with larger diameter, the twist-induced tension, bending and compressive deformation play a more important role compared with that in the smaller bundles.^{11,20,41} On the other hand, as aforementioned the interface force/friction during the pull-out of the multi-walled CNT will rely on the overlapped length when its length is much larger than a crossover length.³⁴ Thus, a further thorough investigation is still anticipated to unveil the scale effect on the mechanical properties of DNT bundles.”

Response to Reviewer 2

Review Comments: *This is an interesting and timely paper on mechanical properties of nanothread bundles.*

Author reply: We are grateful to the reviewer for commenting our work as an interesting and timely study. We also thank the reviewer for her/his constructive comments and suggestions. Below we give our detailed response in a point-to-point manner. The manuscript has also been revised accordingly (see the list of changes.)

Specific Comments:

Comment 1: *First, the precise atomic structure of nanothreads is not yet known - many possibilities have been enumerated as indicated by the references in the introduction. The load transfer between threads, for example, is likely to be sensitive to the precise structure of the constituent threads. Thus this issue should be explored more thoroughly, or else it is not known how widely the results obtained here apply to real threads.*

Author reply: We agree with the reviewer that the load transfer mechanism depends on the precise structure of the nanothreads. The current paper has focused on one type of the nanothreads that contains Stone-Wales transformation defects, which is similar as the structure being discussed in the initial experimental work (Fitzgibbons *et al*, *Nat Mater*, **2015**, 14, 43-47).

According to the geometrical characteristics, the nanothreads can be divided into three categories, including achiral, stiff chiral, and soft chiral. A thorough investigation of all nanothread bundles require a substantial work, and will be carried out in our following paper. However, as discussed in this work, the efficient load transfer in DNT bundle is originated from the strong mechanical interlocking effect, and such interlocking effect is resulted from its zigzag morphology or surface irregularities. With this consideration, the nanothread with highly irregular surface is expected to possesses a high interface load transfer efficiency. In the revision, we have conducted additional calculations to verify this point.

In the revision, we have added the following discussions to ensure a more comprehensive discussion (the corresponding simulation results are included in Supporting Information S9).

*On Page 17-18: “Before concluding, we’d like to mention that there are three types of nanothreads including achiral, stiff chiral and soft chiral (classified according to their geometrical characteristics),¹⁴ and the current work has focused on the DNT with Stone-Wales transformation defects. Above discussions have clearly shown that the effective interface load transfer efficiency for the DNT bundle originates from the strong mechanical interlocking effect, which is resulted from the zigzag morphology or surface irregularities of the constituent DNTs. Therefore, it is expected that nanothreads with highly irregular surface will possesses a high interface load transfer efficiency (a complete comparison of the geometrical structures of the DNTs can be found from the work by Xu *et al.*¹⁴). In this regard, we have selected another two representative DNT bundles, one is the straight DNT – with no SWD and has a smooth surface, and the other one belongs to the soft chiral group, which has a stable spiral structure – with a highly irregular surface (Figure 6b). Following the same simulation settings, we find that the straight DNT has a very small transferred load around 0.25 eV/Å. Whereas, the the spiral nanothread exhibits a high maximum transferred load (~ 3.3 eV/Å) at the early stage of the sliding, which leads to the bond breaking of the structure. This result indicates that the interfacial load can be loaded up to the elastic limit of the soft chiral nanothread. By assuming that the core DNT is under a pure tensile deformation, such maximum transferred load equals to a tensile stress of ~ 35 GPa. Overall, these results have well verified our assumption that smooth surface nanothreads will possess a low interface load transfer efficiency, while the nanothreads with highly irregular surface will have a very high load transfer efficiency. The details are given in Supporting Information (S9).*

Figure 6 (a) Comparisons of the transferred load between DNT bundle and CNT bundles. (b) DNT bundles constructed from the smooth DNT (without SWD), and a spiral soft DNT (bottom image clearly shows the spiral feature of the soft DNT). To avoid the spiral end influence, the core strand is longer than the surrounding DNTs (one-unit length in each end). A shorter bundle has been considered as the transferred load is independent of the overlapped length, with the surrounding strands approximate to 10 nm. The sliding load was imposed on the extrude end of the core DNT.”

Comment 2: *Second, it is unclear how well the inter atomic potential chosen can handle variations in the inter-tube interaction with variations in registry. There are, for example, registry-dependent potentials that indicate that a simple LJ treatment of this interaction is much too smooth (and AIREBO uses a not so different form for this component of the interaction). Furthermore, the accuracy of this potential under transverse compression of the bundle (such as would be seen under torsion) needs to be addressed.*

Author reply: We agree with the reviewer that the selection of potential to describe inter-tube interaction is critical in this study. Actually, both AIREBO and REBO empirical potentials have been reported to reasonably capture the vdW interactions in multi-layer graphene (Zhang *et al*, *Carbon* **2015**, 94, 60-66), multi-wall carbon nanotubes (Xia *et al*, *Phys Rev Lett*, **2007**, 98, 245501), carbon nanotube bundles (Ni and Sinnott, *Surf Sci*, **2001**, 487, 87-96), and hybrid carbon structure (Barzegar *et al*, *Nano Lett*, **2015**, 15, 829-834).

Regarding the transverse compression, AIREBO as well as REBO potentials contains dihedral terms which describe the torsional interactions, and thus enable both potentials to accurately reproduce the elastic properties of carbon systems (e.g., carbon nanotube, graphene), including elastic constants, Poisson’s ratio. Thus, both of them have been widely employed to study the compressive behaviors of carbon systems, such as the compression of CNT bundles (Ni and Sinnott, *Surf Sci*, **2001**, 487, 87-96), torsion of CNT (Jeong *et al*, *J Appl Phys*, **2007**, 101, 084309; Zhang *et al*, *Carbon*, **2010**, 48, 4100-4108), tension-twisting deformation of CNTs (Faria *et al*, *Compos Sci Technol*, **2013**, 74, 211-220). Lennard-Jones potential has been used to model the interwall interaction in multi-walled carbon nanotubes (Z. Xia and W. A. Curtin, *Phys Rev B*, **2004**, 69, 233408), and the pullout force calculated by MD simulation agreed well with the experimental data (M. F. Yu, B. I. Yakobson, and R. S. Ruoff, *J Phys Chem*, **2000**, 104, 8764).

In the revision, following discussions have been added to justify the usage of AIREBO potential.

On Page 19-20: “This potential includes short-range interactions and long range van der Waals interactions, which has been shown to well represent the binding energy and elastic properties of carbon materials. It contains dihedral terms which describe the torsional interactions, and thus enable to accurately reproduce the elastic properties of carbon systems, such as the compression of CNT bundles,⁴⁴ torsion of CNT,^{45,46} tension-twisting deformation of CNTs.⁴⁷ Moreover, it adopts Lennard-Jones term to describe the van der Waals

interactions, which has been reported to reasonably capture the vdW interactions in multi-layer graphene,⁴⁸ multi-wall carbon nanotubes,⁴⁹ carbon nanotube bundles,⁴⁴ and hybrid carbon structure.^{50,}

***Comment 3:** Finally, it is possible for some of the high-symmetry thread structures to have a radial breathing mode, contrary to the assertion made here.*

Author reply: We agree with the reviewer that the high-symmetry thread structures may have a radial breathing mode. In our Supporting Information – S6, we have discussed the radial breathing mode (RBM) in the (10,10)CNT bundles. From the simulation, we can clearly observe the breathing mode of the CNT during the sliding process. Thus, the RBM could also be reproduced for the thread structures if exist. The ultrathin DNT emphasized in this work has a high density of Stone-Wales transformation defects, thus, no radial breathing mode is expected. From our simulation, we also didn't observe the appearance of RBM for the DNT.

Response to Reviewer 3

Review Comments: While the analysis is solid and the manuscript is well written, the lack of experimental validation undermines the potential impact of this work. In fact, it wasn't clear when I first reviewed the abstract that the work is entirely theoretical in nature (especially the second sentence reads "In this work, we find that the recently synthesized ultrathin diamond nanothread not only posses excellent torsional deformation,..."). The conclusion that "the load transfer is independent of overlapped length" is somewhat surprising and (at least for the CNT case) contradicts both simulation results by Yakobson *et al.* and experimental work as reviewed in Behabtu, Natnael, Micah J. Green, and Matteo Pasquali. "Carbon nanotube-based neat fibers." *Nano Today* 3.5 (2008): 24-34.

Author reply: We are grateful to the reviewer for commenting our manuscript as solid and well written.

The main concern on our manuscript from Reviewer 3 is the conclusion that the load transfer is independent of overlapped length contradicts with two previous references. We have reviewed this issue carefully and found that our statement agrees well with most of the theoretical models, simulation results and experimental measurements published previously.

As an example, following we compared the interface load *vs* displacement from our current work (Left, Figure 3 in the manuscript) and the interface stress *vs* displacement during pull-out of multi-walled CNT (Right) as reported by Xia and Curtin [1]. Apparently, both pull-out processes are similar to each other, and the interface load/stress is independent of the overlapped length.

We agree with the reviewer that there is inconsistency with the two mentioned papers. To our understanding, the inconsistency originated from the scale effect. In principle, the pull-out process of the CNT fiber is analogue to the pull-out of the inner CNT from a multi-walled CNT, and the difference lies on the contact area. According to the theoretical analysis [2], there exists a critical crossover length L^* (over 200 nm for CNT), under which the minimum force required to pull out the inner tube is independent of the overlapped length. This length-independent phenomenon has been affirmed from both MD simulations [2-5] and *in situ* measurements [6-9] for the pull-out of multi-walled CNTs with short length. Thus, it is reasonable for us to observe the interface load independent of the overlapped length (as the system in MD simulation is in the order of 10 nm). On the other hand, in the work mentioned by the reviewer, *Carbon*, **2000**, 38, 1675-1680, the authors presented the results of the approximate mean-field based model calculation of the overall strength of a rope. Their results also support that for short CNT, the interface friction is independent on the overlap length. However, for long CNT, the friction will be length dependent. In the other work mentioned by the reviewer, *Nano Today*, **2008**, 3, 24-34, the length of the CNT fiber is in the order

of μm , and there exists defects in the CNT. Therefore, in the suggested work *Nano Today*, **2008**, 3, 24-34, the interface friction is expected to be dependent on the overlapped length (different from that in short CNTs). Thus the results presented in our work don't contradict with those in literatures.

References:

- [1]. Xia, Z.; Curtin, W., *Phys. Rev. B* **2004**, 69 (23), 233408.
- [2]. Huhtala *et al*, *Phys. Rev. B* **2004**, 70 (4), 045404.
- [3]. Kolmogorov, A. N.; Crespi, V. H., *Phys. Rev. Lett.* **2000**, 85 (22), 4727.
- [4]. Tangney, *et al*, *Phys. Rev. Lett.* **2004**, 93 (6), 065503.
- [5]. Li *et al*, *Carbon* **48**, 2934-2940.
- [6]. Cumings, J.; Zettl, A., *Science* **2000**, 289 (5479), 602-604.
- [7]. Kis, *et al*, *Phys. Rev. Lett.* **2006**, 97 (2), 025501.
- [8]. Zhang, *et al*, *Nat. Nanotech.*, **2013**, 8, 912-916.
- [9]. Zhang, *et al*, *Nano Lett*, **2016**, 16, 1367-1374.

In the revision, we have removed the unnecessary statements which emphasize the length independent phenomenon, and added the following discussion to justify this inconsistency issue.

On Page 11: “We should stress that the observed length independence of the transferred load in CNT bundle is consistent with previous results from both MD simulations and *in situ* pull-out of multi-walled CNTs.^{24,30-33} Theoretical studies have shown that there is a strong size scale on the load transfer mechanisms during the pull-out of a multi-walled CNT,³⁴ that is the pull-out force will be independent of the overlapped length when the overall length is much smaller than a crossover length L^* (which is over 200 nm). However, when the overall length is longer than the crossover length, the interfacial friction is length dependent, which is also predicted from the approximate mean-field based model,³⁵ and observed experimentally when the length of the CNT fiber is in the order of μm .³⁶”

List of Changes

(Note: *only major changes are listed below.*)

1. On page 7, we have added discussion about the torsional stiffness of a single DNT and single CNT. We have added one figure (Figure S2) and one section (S2) in the Supporting Information. This is in response to Comment #1 of Reviewer 1.
2. On page 8, we have added discussion about the twist deformation of CNT/DNT bundles. This is in response to Comment #2 of Reviewer 1.
3. On page 7-8, we have added discussion about the torsional rigidity of CNT and DNT bundles. This is in response to Comment #3 of Reviewer 1.
4. On page 8, we have added discussion about the effect of SWD spacing. We have added one figure (Figure S4) and one section (S4) in the Supporting Information. This is in response to Comment #4 of Reviewer 1.
5. On page 16, we have added description in Figure caption. This is in response to Comment #5 of Reviewer 1.
6. On page 8, we have added discussion about the location dependent deformation. This is in response to Comment #6 of Reviewer 1.
7. On page 15, we have added discussion about the size effect. This is in response to Comment #7 of Reviewer 1.
8. On pages 8& 17, we have added two paragraphs to discuss the results of another three CNTs. We have added one figure (Figure 6) in the main manuscript, two figures (Figures S3&S8) and two sections (S3&S8) in the Supporting Information. This is in response to Comment #8 of Reviewer 1.
9. On page 19, we have added discussion. This is in response to Comment #9 of Reviewer 1.
10. On page 9, we have added description of additional calculation result. And on page 15, we have added one paragraph to discuss the size effect. These are in response to Comment #10 of Reviewer 1.
11. On page 17-18, we have added discussion about different DNT bundles. We have added one figure (Figure S9) and one section (S9) in the Supporting Information. These are in response to Comment #1 of Reviewer 2.
12. On page 19-20, we have added discussion about the potential adopted in this study. This is in response to Comment #2 of Reviewer 2.
13. We have removed the unnecessary statements, added two references (now Refs. 35, 36) and added one paragraph on page 11 to compare with previous literatures. This is in response to Comment of Reviewer 3.
14. On page 20, we have updated the Supporting Information.
15. On page 20, we have included our acknowledgement to Dr Tao Liang from Pennsylvania State University for the insightful discussion on the AIREBO potentials.
16. Overall, we have made substantial changes (including new results and more extensive discussions) to the manuscript following the editor's suggestion.

REVIEWERS' COMMENTS:

Reviewer #1 (Remarks to the Author):

The authors have adequately addressed my previous comments. The manuscript has been improved. I appreciate the effort particularly in the modelling of additional diameter nanotubes (e.g., (4,3), (0,8), and (5,5)), which adds rigour to the claims. The additional insight into torsional rigidity is also clarifying. Clearly, there are addition questions arising with this material/system, but they should be left for future studies - this effort lays the groundwork for a novel, emerging system, beyond the extremely numerous "CNT bundle papers".

Reviewer #2 (Remarks to the Author):

The authors have suitably answer my concerns in the original report, except for one thing. My concerns about using AIREBO for phenomena that are sensitive to the VdW interactions between sp² sheets (i.e. the nanotube case) related not to the ability of the AIREBO to reproduce well the overall binding energy arising from VdW, but the variations in binding with changes in the interlayer registry. The authors reply on this point did not explicitly engage with this issue. But this effect should not be an issue for the nanothreads themselves since they are sp³ and "bumpy", and the nanothreads are the main topic here. So all that is required is the addition of a a brief caveat about this effect, and I do not need to see the paper again.

Reviewer #3 (Remarks to the Author):

The authors have adequately addressed my comments. I do not oppose publication.

Response to Reviewer 1

Review Comments: *The authors have adequately addressed my previous comments. The manuscript has been improved. I appreciate the effort particularly in the modelling of additional diameter nanotubes (e.g., (4,3), (0,8), and (5,5)), which adds rigour to the claims. The additional insight into torsional rigidity is also clarifying. Clearly, there are additional questions arising with this material/system, but they should be left for future studies – this effort lays the groundwork for a novel, emerging system, beyond the extremely numerous “CNT bundle papers”.*

Author reply: We are grateful to the reviewer for recommending our work for publication.

Response to Reviewer 2

Review Comments: *The authors have suitably answered my concerns in the original report, except for one thing. My concerns about using AIREBO for phenomena that are sensitive to the VdW interactions between sp^2 sheets (i.e. the nanotube case) related not to the ability of the AIREBO to reproduce well the overall binding energy arising from VdW, but the variations in binding with changes in the interlayer registry. The authors reply on this point did not explicitly engage with this issue. But this effect should not be an issue for the nanothreads themselves since they are sp^3 and “bumpy”, and the nanothreads are the main topic here. So all that is required is the addition of a brief caveat about this effect, and I do not need to see the paper again.*

Author reply: We are grateful to the reviewer for the further explanation regarding the AIREBO shortcomings while describing the binding energy with changes in the sp^2 interlayer registry. As suggested, we have added the following sentence to further clarify this issue in the revision:

On Page 17-18: “To note that the Lennard-Jones term is reported to be deficient in describing the sp^2 interlayer interactions with changes in the interlayer registry,³⁰ which however will not affect our results as this work focused on DNTs and they are sp^3 carbon structures.”

Response to Reviewer 3

Review Comments: *The authors have adequately addressed my comments. I do not oppose publication.*

Author reply: We thank the reviewer for the recommendation for publication.